# Successful Clinical Avoidance of Colorectal Anastomotic Leakage through Local Decontamination

**DOI:** 10.3390/antibiotics13010079

**Published:** 2024-01-15

**Authors:** Gerhard Ernst Steyer, Markus Puchinger, Johann Pfeifer

**Affiliations:** 1Division of General, Visceral and Transplant Surgery, Medical University of Graz, Auenbruggerplatz 5, 8036 Graz, Austria; gerhard@steyer.at; 2Doctoral School of Lifestyle-Related Diseases, Medical University of Graz, Neue Stiftingtalstraße 6, 8010 Graz, Austria; 3Medical Engineering and Computing, Department of Surgery, Medical University of Graz, Auenbruggerplatz 29, 8036 Graz, Austria; markus.puchinger@medunigraz.at

**Keywords:** local decontamination, anastomotic leak, gentamicin, antibiotics, colorectal surgery, *P. aeruginosa*, *E. faecalis*

## Abstract

Aim: An anastomotic leak is an unpredictable postoperative complication during recovery from colorectal surgery that may require a re-operation. Potentially pathogenic bacteria like Pseudomonas (and Enterococcus) contribute to the pathogenesis of an anastomotic leak through their capacity to degrade collagen and to activate tissue matrix metalloprotease-9 in host intestinal tissues. The microbiome, therefore, is the key to preventing an anastomotic leak after colorectal surgery. The aim of this trial was to investigate whether perioperative selective decontamination with a new mixture of locally acting antibiotics specially designed against *Pseudomonas aeruginosa* and *Enterococcus faecalis* can reduce or even stop early symptomatic leakage. Method: All hospitalized patients in our University Clinic undergoing colorectal surgery with a left-sided anastomosis were included as two groups; patients in the intervention group received polymyxin B, gentamicin and vancomycin every six hours for five postoperative days and those in the control group did not receive such an intervention. An anastomotic leak was defined as a clinically obvious defect of the intestinal wall integrity at the colorectal anastomosis site (including suture) that leads to a communication between the intra- and extraluminal compartments, requiring a re-do surgery within seven postoperative days. Results: Between February 2017 and May 2023, a total of 301 patients (median age of 63 years) were analyzed. An anastomotic leak was observed in 11 patients in the control group (n = 152), but in no patients in the intervention group (n = 149); this difference was highly significant. Conclusion: The antibiotic mixture (with polymyxin B, gentamicin and vancomycin) used for local decontamination in our study stopped the occurrence of anastomotic leaks completely. According to the definition of anastomotic leak, no further surgery was required after local perioperative decontamination.

## 1. Introduction

Anastomotic leak (AL) is an unpredictable postoperative complication during recovery from colorectal surgery that may require re-operation and is associated with high mortality and morbidity rates, but the underlying mechanism is still not (completely) understood. Generations of surgeons have been indoctrinated for decades to learn the current technique of how to create an anastomosis [1] because surgery is traditionally mainly a “manual craft” (predominantly an “art”, and surgeons are therefore “artists”) rather than a “medical science”. To date, interdisciplinary teams have found that the “microbiome is the key to preventing AL in colorectal surgery, representing a critical missing piece in this puzzle—it modulates the innate immune response to anastomotic wound healing” [2,3].

For decades, surgeons have tried to find out the reasons that cause AL; a comprehensive but unsuccessful discussion in the academic literature and during medical conferences was initiated to find the proper surgical technique [4,5], while “patient-related factors” were also intensively investigated.

Now we know that bacteria (nosocomial microbes from the hospital and microbiota from the patient’s own gut that can shift perioperatively to a pathogenic flora) are mainly involved in a patient’s own AL [6,7]. A shift to a pathogenic microbiome can also happen due to medication or radiation [8,9] or depend on age, sex and hormonal situation [10,11,12]. Potentially pathogenic bacteria like Pseudomonas (and Enterococcus), which occur in small amounts in the commensal gut flora, “contributes to the pathogenesis of anastomotic leak through its capacity to degrade collagen and to activate tissue matrix metalloprotease-9 (MMP9) in host intestinal tissues” [13,14] at the anastomosis; *Bacillus subtilis* is also involved and has been investigated [15]. 

In recently published reviews [16,17], relevant strategies to eradicate multidrug-resistant (MDR) bacteria have been discussed.

## 2. Method

### 2.1. Defining Graded AL after Colorectal Resection and the Observation Time in This Study

In the medical literature, which represents (besides empirical knowledge) our conventional medical wisdom (in part), there is a surprising lack of a standardized definition of AL (even though AL represents a major feared complication after gastrointestinal surgery, resulting in a widely reported [9] variation from 4 to 20%, for example), based on the heterogeneity in the AL definition in many studies [18]. Bruce identified [19] a total of 56 different definitions of anastomotic leak in 97 reviewed studies at three sites: “lower gastrointestinal (29 definitions), hepatopancreaticobiliary (14 definitions) and upper gastrointestinal (13 definitions)”. The rate of AL has failed to substantially improve over the years. Therefore, our first aim in the present study was to define AL itself for the study protocol submitted to the ethics committee, its proposed feasible cause(s), the entity (early vs. long) and, consequently, the endpoint and the duration of the observation time. 

The temporal occurrence of AL during recovery from colorectal surgery also has to be defined for the protocol because it is well known that symptomatic anastomotic leakage that requires re-do surgery following surgical resection can occur during the initial hospital stay (up to 7 postoperative days) or later, after hospital discharge. The point in time of AL is very important [20] with respect to the severity and management of this complication. Late AL is defined [21,22,23,24] as leakage that occurs after 30 postoperative days, but this is a rare event [25] (occurring in less than 4% of colorectal cases in one study [26]). By contrast, early AL is associated with severe peritonitis, emergency re-do surgery and increased mortality, with a median occurrence time of 5–6 postoperative days [20,27]. We follow the study results from Floodeen [28] in that early and late symptomatic leakage may be viewed as different entities. 

The severity of anastomotic leakage should be graded according to the impact on clinical management [18]. Grade A: anastomotic leakage results in no change in the patient’s management. Grade B: leakage requires conservative management but no re-do surgery. Grade C: anastomotic leakage requires re-do surgery. In this study protocol, we defined Grade C as the end of the investigation. 

AL can be defined primarily as a leak of luminal contents from a surgical join between two hollow viscera [29]. In our study protocol, we add on two more clinically important factors: point of time [20] of AL and, secondly, the requirement of re-do surgical intervention (=severity of AL). Therefore, we determined AL in daily, routine procedures at a university hospital (with a high rate of emergency surgery and not an academic inclusion criteria of “only elective resections”), which was modified from the literature [18,20,21] as follows: AL is a clinically obvious defect of the intestinal wall integrity at the colorectal anastomosis site (including suture) that leads to a communication between the intra- and extraluminal compartments, requiring re-do surgery within 7 postoperative days.

### 2.2. Patients

We included all consecutive patients (older than 18 years of age) from our clinic who required left-sided colorectal anastomosis; that means not only elective but also urgent patients, representing the daily routine at a university hospital. 

We did not include pregnant females (because of the antibiotic intervention) or persons with known antibiotic allergy.

No restrictions were applied regarding gender.

We compared an intervention group with a retrospective control group. 

### 2.3. Design

We compared an intervention group and a retrospective control group (without local antibiotic decontamination) in a mono-center study of AL after left-sided colorectal anastomosis. In the peri- and postoperative period, the clinical decisions were identical in both groups. If, in the emerging situation, a diverting stoma was required, topical administration of the study drug was given transanally.

Day of inclusion (T0) was the day of surgery. The intervention period followed from T0 to T5. On day T7, the observation time ended (see definition of AL).

### 2.4. Antibiotic Intervention

In the intervention group, patients received (in addition to the standard antibiotic prophylaxis starting preoperatively, which was an i.v. of piperacillin/tazobactam—a broad-spectrum ß-Lactam antibiotic that can act via penetration in Gram-negative bacteria, as well as against *P. aeruginosa*, but when used alone, it lacks strong activity against the Gram-positive pathogens) an antibiotic mixture 4 times a day for 5 consecutive days after surgery, beginning the day before surgery in elective patients and as soon as possible in emergency patients. With respect to previous studies, we chose a mixture (PGV) that could act against Gram-positive and Gram-negative pathogens (like *P. aeruginosa* or *E. faecalis*, respectively): polymyxin B (100 mg), gentamicin (80 mg) and vancomycin (125 mg). The PGV medication was chosen by our study group according to the approved study protocol and was then prepared by our pharmacy at the Medical University of Graz and filled in capsules.

During the inclusion period, we had a slight delay in our timetable due to COVID-19 (SARS-CoV-2); there was no change in the suture material (polydioxanone 4-0 or 5-0), stapling device or technical management of gut surgery. Clinical data were retrospectively obtained from our internal documentation system (MEDOCS). 

### 2.5. Statistics

Patients’ characteristics are presented as medians and expressed as a percentage. For the comparisons, we used the Mann–Whitney U test for continuous data and Fisher’s exact test for binary data. A *p* value < 0.05 was considered statistically significant, and a *p* value < 0.001 highly significant.

Statistical analyses were carried out using IBM SPSS Statistics v29 (IBM Corp., Armonk, NY, USA).

## 3. Results

### 3.1. Results of Interventions for Preventing AL in the Previous Literature

We found in our literature search three studies using PT (=polymyxin B (or E) and tobramycin) for decontamination [30,31,32,33], resulting in a minor (in two studies, not significant) reduction in AL (from 9.7% to 6.6%). Another study [34] had no control group and used PG (=polymycin B and gentamicin) and reported AL in 5.8%.

Three other studies [35,36,37] used PTV (=polymyxin B (or E), tobramycin and vancomycin but no gentamicin) and reported, respectively, a reduction in AL from 14.9% to 6.5%, 20% to 5% and 10.6% to 2.9%. Also, these three studies investigated elective patients only.

### 3.2. Patients’ Demographic Data

The two cohorts did not differ in baseline parameters (see Table 1); CG and IG were comparable with respect to the patients’ characteristics.

In the previous literature, we saw a remarkable reduction in AL through local decontamination, but our antibiotic mixture was used for the first time in this trial. 

### 3.3. Results of the Intervention

In the CG (n = 152), we found 11 (=7.23%) necessary re-do operations vs. none in the IG (n = 149, with the new local decontamination with PGV); this result is highly significant and in consensus with medical studies and the literature.

## 4. Discussion and Conclusions

### 4.1. Requirements

Anastomosis must heal in the presence of fecal contamination [38].Microbiota, perioperatively, can shift to a pathogenic flora and contribute to the pathogenesis of AL through its capacity to activate MMP9 and degrade collagen in host intestinal tissue [13].A mixture of locally acting antibiotics—specially designed against *Pseudomonas aeruginosa* and *Enterococcus faecalis*—can reduce AL [36].

### 4.2. Discussion of Former Findings in the Literature

To reach our aim (=to reduce AL in daily, routine patients undergoing left-sided colorectal surgery in our hospital which has a high rate of emergency surgery), we had to find the proper intervention drug. The result of our literature check was the following:, after a historical pioneer milestone case report series from the German surgeon Martin Wilhelm von Mandt [39] (University of Greifswald), and after several animal studies [38,40,41,42,43,44], patient studies [45,46,47,48,49] followed, investigating different antibiotics to reduce AL in elective patients (one [47] without success in significant reduction in AL; one [49] a meta-analysis of thirteen studies, with very different antibiotics), there were four published studies (three studies [30,33,34] and the underpowered SELECT-trial [31,32]) that were using PT/PG (PT: polymyxin/tobramycin; PG: polymyxin/gentamycin but no vancomycin) and amphotericin B (an antimycotic acting substance) for decontamination, all showing a slight reduction in AL for elective patients. “Gut microbiota can play a critical role in the healing process of anastomotic tissue and alterations in its composition may be largely to blame for anastomotic insufficiency”.

In 1997, Hans Martin Schardey (Agatharied Academic Teaching Hospital of the Ludwig-Maximilians-University Munich, Germany) published his milestone trial [35], using PTV (PTV: polymyxin/tobramycin/vancomycin) and amphotericin B for locally effective acting decontamination in patients’ gut with a remarkable reduction in AL. Based on the excellent evidence, perioperative local decontamination was implemented at our surgical institution at the University of Graz; the occurrence of AL with regard to this change was an aim of our trial.

The next study by the group of Schardey [36] was cancelled after an interim analysis because of a death in the control group, not in the group using the investigation drug. This setting and concept were also investigated by a group of German researchers at the University of Dresden [37] for elective patients only, and it resulted in a reduction in AL in the decontamination group.

A human body exists in symbiosis with its microbiota, which can be defined as the ecosystem of the body (influencing, e.g., the oral microbiome [10], the skin and the immune system [50], the gut and vagina [11] and the prostate [51] during the perinatal period and in old age [10]). A microbial dysbiosis contributes to “diseases” and consists of changes in the microbial metabolism; this influences the regulation of inflammation, e.g., in the case of AL, a change in the release of metabolites influences the gut barrier and wound healing after gut surgery. 

It has been well known for a long time that microbiota can modulate the restitution during a postoperative period, not only by influencing the immune system but also by influencing metabolites. “Bacterial metabolites can be either degraded or absorbed depending on competitive microbes” [52], and “surgical injury … can shift the phenotype of a potentially pathogen from innocuous colonizer to invasive and virulent pathogen” [13]. The release of host stress factors during surgery can activate bacterial virulence genes. Until now, P. aeruginosa and E. faecalis have been identified and investigated [13,53] to shift to a pathogen phenotype and act after gut surgery via the activation of MMP9, followed by the degradation of collagen. An “adequate anastomotic healing requires collagen deposition and remodeling through post-translational modification” [9].

To date, we have identified four published studies [30,31,32,33,34] in the medical literature on AL in patients who received an antibiotic mixture that is known under the name SDD (Selective Decontamination in Digestive tract) and is helpful for avoiding respiratory infections in emergency care units, especially against *P. aeruginosa*. SDD contains polymyxin B and tobramycin (an aminoglycoside antibiotic). In AL patients, amphotericin B (a polyene-antimycotic agent) was always added to the mixture; this intervention results in a moderate reduction in AL. Two trials had to be stopped after an interim analysis because those studies were statistically underpowered. 

A modification of the agents for SDD was used in three other trials [35,36,37]. Even here, one trial was stopped (because of an unexpected side effect in the control group); also, these trials resulted in a remarkable reduction in AL. 

Based on these, we designed a study [54] in elective and emergency patients undergoing colon surgery (representing the routine cases in a university hospital). Therefore, we used P + G (P: polymyxin; G: gentamicin) for the decontamination of the Gram-negative bacteria (like *P. aeruginosa* and *E. faecalis*). For the decontamination of Gram-positive bacteria (like *C. difficile* and its toxins [55,56]), we added the well-investigated antibiotic vancomycin (but no antimycotic like amphotericin B), resulting in zero AL (vs. eleven cases in the control group). The only difference concerning treatment in our two groups was the use of PGV in the treatment group because both groups received systemic piperacillin/tazobactam following the standard protocol in our clinic [57]. 

### 4.3. Discussion of the Hypothesis of This Study

In 1987, Unertl [58] published his study on the “prevention of colonization and respiratory infections in long-term ventilated patients by local antimicrobial prophylaxis”. He investigated a mixture of polymyxin B and gentamicin because “gram-negative rods, especially *Pseudomonas aeruginosa* …… were found to be sensitive to at least one of the two agents, …… gentamicin covered more than 90% of the isolates of *Enterobacteriaceae* ……, but only 80% of the isolates of *P. aeruginosa*. Both agents exert a high and prolonged local activity when given orally, show no systemic toxicity as they are not absorbed through intact mucosa and are well tolerated. Cross-resistance between polymyxin B and other antimicrobial agents is uncommon, and amikacin could be expected to be a suitable substitute if resistance against gentamicin developed”. We decided in the protocol of our study to investigate this mixture (polymyxin B and gentamicin) combined with vancomycin, a glycopeptide antibiotic, covering Gram-positive microbes like *C. difficile*, MRSA and *E. faecalis*. We wanted to kill mainly *E. faecalis* and *P. aeruginosa* (because of their property to activate MMP9 and consecutively collagenase resulting in degradation of collagen at the anastomosis), and therefore we did not add amphotericin B. 

Our study hypothesis, that the occurrence of AL can be reduced through a locally acting unabsorbable antibiotic mixture, has been completely validated and we showed, indirectly, that MMP9-activating microbes are involved in the disturbed healing process of an anastomotic leak.

### 4.4. Discussion of the Intervention in This Study

By using the described intervention with a mixture of polymyxin B, gentamicin and vancomycin, we stopped AL in our study completely compared to the control group. This combination of antibiotics was used for the first time in this study to prevent AL—and the game was worth the candle. 

A strength of this study is that the intervention worked beside the high rate of emergency surgery in our clinic. Future studies should be randomized.

We have been able to successfully fight nosocomial hospital microbes with reserve antibiotics. Until now, three main bacteria that cause AL have been investigated: *P. aeruginosa*, *E. faecalis* and *B. subtilis*. The antibiotic mixture therefore includes polymyxin B (a polypeptide antibiotic) to fight Gram-negative bacteria like *P. aeruginosa*, and gentamycin (instead of tobramycin, fighting *P. aeruginosa*), an aminoglycoside antibiotic, to fight *P. aeruginosa* and *E. faecalis*, and, furthermore, vancomycin (a glycopeptide antibiotic) to fight Gram-positive bacteria like *E. faecalis*; however, this study did not include the antimycotic acting amphotericin B (a polyene antimycotic acting substance). *Bacillus subtilis* was incriminated to cause AL [15] and is also fought by vancomycin. 

Patients additionally received a piperacillin i.v., a systemic antibiotic prophylaxis. Piperacillin must be administered parenterally, as after oral administration, only minimal absorption occurs. It is well known [59] that, if the broad-spectrum penicillin piperacillin is given “in combination with aminoglycoside antibiotics” (like gentamicin in our study), “additive and synergistic increases in activity can often be achieved against Enterobacteriaceae and especially against *Pseudomonas aeruginosa*”.

The decontamination medication (PGV) was prepared by our institutional pharmacy. 

### 4.5. Discussion of Results of This Study

In our study, ASA score did not significantly influence the leak rate; that is in consensus with the medical literature [60], and risk factors were not an aim of our investigation.

The PGV intervention in our study showed that AL can be stopped successfully (or at least reduced remarkably), but this means that selectively killing bacteria like *P. aeruginosa* and *E. faecalis* with a topic-acting antibiotic mixture is also indirect proof that these killer microbes are strongly involved in the development of AL. 

A re-operation is always associated with different risks. A lifelong stoma can be a consequence, as can increased mortality or morbidity [61].

In Section 3.1 (results of interventions for preventing AL in the previous literature), we compared PGV with available data from other published studies like PTV (three references), PT (three references) and PG (one reference) and other antibiotic regimes. We found PGV to be superior.

The method used in our study seems to be a possible, well-tolerated, successful way for both elective and emergency patients requiring colorectal surgery to avoid AL.

### 4.6. Implication on Future Practice and Research

By using a new antibiotic mixture (PGV) for perioperative decontamination, it was possible to stop AL in elective and emergency patients.

Future (randomized) studies should investigate whether PGV has to be administered routinely to all patients or just in selected patients.

In a systematic review and meta-analysis, a Canadian Study Group [62] “detected no relation between the use of SDD … and the development of antimicrobial resistance in pathogens in patients in the intensive care units”. 

Long-term effects like antimicrobial resistance should be investigated carefully. Also, the question remains if antimycotics (like Amphotericin) have to be added to the antimicrobial decontamination to avoid mycotic overgrowth.

A dose-finding research study could help to reduce antibiotics use (with respect to BMI). 

### 4.7. Reducing Costs

It is not primarily a medical issue to reduce costs, but for AL, it has been shown in some publications concerning AL that decontamination can remarkably reduce the rate of re-operations and, in this way, healthcare costs [35,37].

Some bookkeepers—from a patient’s point of view—also take increasing costs into consideration and perform an “economic evaluation from a societal perspective as a cost-effectiveness and cost-utility analysis” [31,32]. Ultimately, and not surprisingly, number crunchers recently presented “the first cost-analysis study of AL after colorectal surgery” [63] in colorful pictures, and “concluded” that “the appearance of AL generates a considerable increase in the consumption of health resources, mainly due to an increase in hospital stays” and “the more complex the AL, the higher the cost associated with its treatment”—cost analyses on this topic have already been completed and are therefore obviously not an aim of our investigation.

## Figures and Tables

**Table 1 antibiotics-13-00079-t001:** Patients’ characteristics. IG = intervention group; CG = control group; No. = number of patients; BMI = Body Mass Index; ASA score = American Society of Anesthesiology Classification System; AL = anastomotic leakage.

	IG n = 149		CG n = 152		
	No. (%)	Median	No. (%)	Median	*p* Value
Gender					
Male	80 (53.69%)		72 (47.36%)		
					0.163
Female	69 (46.30%)		80 (52.63%)		
Age		63		63	0.405
BMI		25		25	0.574
ASA					
I	14		12		0.398
II	66		63		0.351
III	52		59		0.279
IV	17		18		0.525
AL					
Redo-surgery	0.0		11 (=7.23 %)		<0.001

## Data Availability

Clinical data were retrospectively obtained from our internal documentation system (MEDOX). More information is available in a dissertation [54] by the first author on this topic; these data have not been published elsewhere.

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
