# Peer review of "Successful Clinical Avoidance of Colorectal Anastomotic Leakage through Local Decontamination"

_antibiotics, 2024, doi:10.3390/antibiotics13010079_

Round 1
Reviewer 1 Report
Comments and Suggestions for Authors
The authors correctly point out that the mechanism of action of PGV in preventing AL is likely due to its ability to reduce the number of pathogenic bacteria in the gut, particularly Pseudomonas aeruginosa and Enterococcus faecalis. These bacteria have been implicated in the pathogenesis of AL by activating MMP9 and degrading collagen at the anastomosis site.
however I have some consideration as minor critcisms
- The study did not assess the long-term effects of PGV on patient outcomes, Please add something on this in discussion
- The study did not compare PGV to other antibiotic regimens for preventing AL, please add some on this in discussion
- Introduction should be enriched by including some other review or metanalisis on the need to treat colonized patients with CRE bacteria as (10.1017/ice.2021.492. or 10.1016/j.jiph.2023.01.009)
- Expand on the mechanism of action of PGV in preventing AL. The authors could provide more detail on the specific mechanisms by which PGV reduces the number of pathogenic bacteria in the gut and how this contributes to the prevention of AL.
- Discuss the potential for PGV to reduce the development of antibiotic resistance. The use of broad-spectrum antibiotics, such as PGV, can increase the risk of antibiotic resistance. The authors could discuss the potential for PGV to reduce this risk by targeting specific pathogenic bacteria.
- Explore the cost-effectiveness of PGV compared to other methods of preventing AL if it possibile. Particularly, the authors could conduct a cost-effectiveness analysis to compare the cost of administering PGV to the cost of other methods, such as reoperation and prolonged hospital stays.
Author Response
Final Cover letter antibiotics-2803235 revisions
Successful Clinical Avoidance of Colorectal Anastomotic Leakage by Local Decontamination
The point by point answers to all comments of the reviewers are stated below. We would like to thank the reviewers for their comments which help in improving the manuscript.
Reviewer 1: Comments and Suggestions for Authors
The authors correctly point out that the mechanism of action of PGV in preventing AL is likely due to its ability to reduce the number of pathogenic bacteria in the gut, particularly Pseudomonas aeruginosa and Enterococcus faecalis. These bacteria have been implicated in the pathogenesis of AL by activating MMP9 and degrading collagen at the anastomosis site.
However, I have some consideration as minor criticisms:
- The study did not assess the long-term effects of PGV on patient outcomes, Please add something on this in discussion
Done, we add in discussion:
In the protocol of the study we did not assess long-term effects of PGV on patients´ outcomes, but from literature it is well known, that PGV is acting selectively against some microbes like P. aeruginosa and that the microbiota in the gut can be restored quickly after an antibiotic treatment. Studies on antibiotic resistance have been done [DANEMAN 2013].
- The study did not compare PGV to other antibiotic regimens for preventing AL, please add some on this in discussion
Done, and we add in discussion: In chapter 3.1 (results of interventions in previous literature) we compared PGV with available data from other published studies like PTV (3 references), PT (3 references) and PG (1 reference) and other antibiotic regimes. We found PGV to be superior.
- Introduction should be enriched by including some other review or metanalysis on the need to treat colonized patients with CRE bacteria as (10.1017/ice.2021.492. or10.1016/j.jiph.2023.01.009)
Done, we added in Introduction:
- Roson-Calero N, Ballesté-Delpierre C, Fernández J, Vila J. Insights on Current Strategies to Decolonize the Gut from Multidrug-Resistant Bacteria: Pros and Cons. Antibiotics (Basel). 2023 Jun 19;12(6):1074. doi: 10.3390/antibiotics12061074. PMID: 37370393; PMCID: PMC10295446.
- Silvestri L, van Saene HK, Weir I, Gullo A. Survival benefit of the full selective digestive decontamination regimen. J Crit Care. 2009 Sep;24(3):474.e7-14. doi: 10.1016/j.jcrc.2008.11.005. Epub 2009 Feb 12. PMID: 19327325.
- Expand on the mechanism of action of PGV in preventing AL. The authors could provide more detail on the specific mechanisms by which PGV reduces the number of pathogenic bacteria in the gut and how this contributes to the prevention of AL.
We described, that the underlying mechanism of action of PGV in preventing AL is likely due to its ability to reduce the number of pathogenic bacteria in the gut, particularly Pseudomonas aeruginosa and Enterococcus faecalis; these bacteria have been implicated in the pathogenesis of AL by activating MMP9 and degrading collagen at the anastomosis site.-
However, to add the mechanism of action of PGV in preventing AL would increase a lot the length of the manuscript and this is not the main purpose of this perspective.
- Discuss the potential for PGV to reduce the development of antibiotic resistance. The use of broad-spectrum antibiotics, such as PGV, can increase the risk of antibiotic resistance. The authors could discuss the potential for PGV to reduce this risk by targeting specific pathogenic bacteria.
We add in the chapter “Discussion (4.7)”:
In a systematic review and meta-analysis a Canadian Study Group [DANEMAN 2013] “detected no relation between the use of SDD … and the development of antimicrobial resistance in pathogens in patients in the intensive care units”.
- and the reference:
Daneman N, Sarwar S, Fowler RA, Cuthbertson BH; SuDDICU Canadian Study Group. Effect of selective decontamination on antimicrobial resistance in intensive care units: a systematic review and meta-analysis. Lancet Infect Dis. 2013 Apr;13(4):328-41. doi: 10.1016/S1473-3099(12)70322-5. Epub 2013 Jan 25. PMID: 23352693.
- Explore the cost-effectiveness of PGV compared to other methods of preventing AL if it is possibile. Particularly, the authors could conduct a cost-effectiveness analysis to compare the cost of administering PGV to the cost of other methods, such as reoperation and prolonged hospital stays.
We mentioned that cost-effectiveness analysis is available from other studies in Europe.
Reducing occurrence of AL can shorten the stay in hospital, redo-surgery increases the cost remarkable. Cost-effectiveness is not a medical issue and was not the aim of our study.
Kind regards.
Reviewer 2 Report
Comments and Suggestions for Authors
Abstract: I would avoid abbreviations in the abstract
Line 14 why the brackets? - “contribute
Line 16 wether - whether
Line 19 What do you mean by „continuous patients”? You mean hospitalized, inpatient?
Lines 31-34 would plase to the end of the Discussion
Line 35 Keywords should be more compressed
References 29 and 30 redundant and differ from Good Clinical Practice guidelines
Line 49 bugs – rename it: microbs …
Line 50 and line 67 Why do you write: (!) ?
There is a contradiction: Line 61 observation time (endpoint) versus Line 90 „Grade C as the endpoint”
Line 109 Here or somewhere alse please describe the method, how you included patients data retrospectively regarding detection or selection bias
Lines 100-111 Patients
- here please describe patients inculsion and exclusion cirteria, age, gender, underlying disease, elective or urgent OP …
- Line 101-103 is a description of the aim of the study – belongs to the Introduction section
- Line 104-106 belongs to Lines 294-296 Institutional Review Board Statement
- Lines 108-111 does not belong to Patients part, better to 2.4
- Lines 110-111: „The decontamination medication (PGV) was prepared from our institutional pharmacy” - belongs to the Intervention section
Line 112 2.3. Design – please give a detailed description of the design here and no results (IG; n=149) and (CG; n=152) yet. Please insert a CONSORT 2010 Flow-diagram. Please define here in this section the exact times of inclusion, intervention, endpoint. For example: T0 = day of inclusion, T0 to T1 = 7 days of intervention, thus T1= T0 + 7 days = endpoint
Lines 115-117: „In the peri- and post-operative period the clinical decisions were identical in both groups. If in the emerging situation a diverting stoma was required, topical administration of the study drug was given transanally” – this belongs to the Intervention section: 2.4
Line 119 2.4. Study drug - would rename it: intervention …
Line 139: 3. Results - Would structure results in subpoints:
Results of interventions for CAL in previous literature
Patients demographic data
Results of the intervention (PVD)
Line 136: „(table 2)” – belongs to the Results section
Line 142 table 2 has to be reorganised, not to read if published as PDF
Lines 276 and 286 and 59: if you mention QoL as an outcome/result of your study, you should underline this with more literature and facts: did you measure it? Did you make any statistics on it?
Discussion section: I would re-structure the Discussion section along: former literature findings, your hypothesis, aims, intervention, results, implication on future practice and research. Now you repeat aims, intervention, results in text form in a mixed up order with unnecessary repetitions.
Please mention weeknesses (bias …) and strength of your investigation.
Line 291-293 This trial was investigator initiated and investigator driven and designed as a doctoral thesis at the Medical University of Graz (Doctoral 292 School of Lifestyle-related Diseases) by G.E.S. under the supervision of J.P. and M.P. – not necessary – this text does not belong here, more in the Authors contribution or Acknowledgment part
Comments on the Quality of English LanguageCarefull typo and scientific English language revision is needed.
Author Response
Final Cover letter antibiotics-2803235 revisions
Successful Clinical Avoidance of Colorectal Anastomotic Leakage by Local Decontamination
The point by point answers to all comments of the reviewers are stated below. We would like to thank the reviewers for their comments which help in improving the manuscript.
Reviewer 2: Comments and Suggestions for Authors
- Abstract: I would avoid abbreviations in the abstract
Done, no abbreviations.
- Line 14 why the brackets? - “contribute
Done, no brackets.
- Line 16 wether - whether
Done, typo repaired.
- Line 19 What do you mean by „continuous patients”? You mean hospitalized, inpatient?
Yes, done.
- Lines 31-34 would place to the end of the Discussion
Done, replaced.
- Line 35 Keywords should be more compressed
Done.
- References 29 and 30 redundant and differ from Good Clinical Practice guidelines
Ref. 29 + 30 are cancelled
- Line 49 bugs – rename it: microbs …
Done.
- Line 50 and line 67 Why do you write: (!) ?
Done, cancelled both (!).
- There is a contradiction: Line 61 observation time (endpoint) versus Line 90 „Grade C as the endpoint”
Intervention T0-T5 (with PGV). Cancelled “(endpoint)“ in line 61, occurence of Grade C (requires redo-surgery and) is he end of observation time.
- Line 109 Here or somewhere alse please describe the method, how you included patients data retrospectively regarding detection or selection bias
Done, it is an observational study (without randomization).
- Lines 100-111 Patients
here please describe patients inclusion and exclusion criteria, age, gender, underlying disease, elective or urgent OP …
Done.
- Line 101-103 is a description of the aim of the study – belongs to the Introduction section
Done, replaced.
- Line 104-106 belongs to Lines 294-296 Institutional Review Board Statement
Done, replaced.
- Lines 108-111 does not belong to Patients part, better to 2.4
Done, replaced to 2.4
- Lines 110-111: „The decontamination medication (PGV) was prepared from our institutional pharmacy” - belongs to the Intervention section
Done, replaced.
- Line 112 3. Design – please give a detailed description of the design here and no results (IG; n=149) and (CG; n=152) yet. Please insert a CONSORT 2010 Flow-diagram.
We performed an observational study (no randomization) and included all consecutive patients.-
Please define here in this section the exact times of inclusion, intervention, endpoint. For example: T0 = day of inclusion, T0 to T1 = 7 days of intervention, thus T1= T0 + 7 days = endpoint
Done, T0-T5 = days of intervention. Day of inclusion (T0) was the day of surgery. The intervention period followed from T0 to T5. On day T7 observation time ended (see definition of AL).
- Lines 115-117: „In the peri- and post-operative period the clinical decisions were identical in both groups. If in the emerging situation a diverting stoma was required, topical administration of the study drug was given transanally” – this belongs to the Intervention section: 2.4
Done, replaced to 2.3.
- Line 119 2.4. Study drug - would rename it: intervention …
Done.
- Line 139: Results - Would structure results in subpoints:
- Results of interventions for CAL in previous literature
- Patients demographic data
- Results of the intervention (PVD)
Done, we used this structure.
- Line 136: „(table 2)” – belongs to the Results section
Done, replaced.
- Line 142 table 2 has to be reorganised, not to read if published as PDF
This table we cancelled, but results are given in 3.1.
- Lines 276 and 286 and 59: if you mention QoL as an outcome/result of your study, you should underline this with more literature and facts: did you measure it? Did you make any statistics on it?
We cancelled QoL and do not address QoL.
- Discussion section: I would re-structure the Discussion section along: former literature findings, your hypothesis, aims, intervention, results, implication on future practice and research. Now you repeat aims, intervention, results in text form in a mixed up order with unnecessary repetitions.
Done, re-structured 4.1 till 4.6.
- Please mention weaknesses (bias …) and strength of your investigation.
Done, in 4.3. (works also in urgent patients. Future studies should be randomized.) and 4.6. (a dose finding research could help to reduce antibiotics - with respect to BMI).
- Line 291-293 This trial was investigator initiated and investigator driven and designed as a doctoral thesis at the Medical University of Graz (Doctoral School of Lifestyle-related Diseases) by G.E.S. under the supervision of J.P. and M.P. – not necessary – this text does not belong here, more in the Authors contribution or Acknowledgment part
Done, replaced to “Funding“.
- Comments on the Quality of English Language:
Carefull typo and scientific English language revision is needed.
Done, language revision by a native speaker.
Kind regards.
Round 2
Reviewer 2 Report
Comments and Suggestions for Authors
Still some typo failures, so please look up the whole document.
line 129: contol group
line 172 and 305: 45/46 45+46
lines 174-175: please revise: there is an author´s memo: in future studies add (!) Vancomycin - that’s what we did in our study, resulting in zero AL! All these studies investigated elective patients only.
Comments on the Quality of English LanguageStill some typo failures, so please look up the whole document.
line 129: contol group
line 172 and 305: 45/46 45+46
lines 174-175: please revise: there is an author´s memo: in future studies add (!) Vancomycin - that’s what we did in our study, resulting in zero AL! All these studies investigated elective patients only.
Author Response
We highlighted our changes in the revised text.
Comments and Suggestions for Authors
- Still some typo failures, so please look up the whole document.
Done. Additionally we used a Spell Checking Programm.
- line 129: contol group
Done. control
- line 172 and 305: 45/46 45+46
Done. 45, 46
- lines 174-175: please revise: there is an author´s memo: in future studies add (!) Vancomycin - that’s what we did in our study, resulting in zero AL! All these studies investigated elective patients only.
Done. We cancelled this statement.
Comments on the Quality of English Language
Still some typo failures, so please look up the whole document.
Done by a Spell Checking Programm.
- line 129: contol group
Done. control
- line 172 and 305: 45/46 45+46
Done. 45, 46
- lines 174-175: please revise: there is an author´s memo: in future studies add (!) Vancomycin - that’s what we did in our study, resulting in zero AL! All these studies investigated elective patients only.
Done. We cancelled this statement.
Thank you